# Scale-Up Cultivation of *Phaeodactylum tricornutum* to Produce Biocrude by Hydrothermal Liquefaction

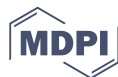

**Irene Megía-Hervás** [1,2], **Alejandra Sánchez-Bayo** [1] , **Luis Fernando Bautista** [3] ,
**Victoria Morales** [3] , **Federico G. Witt-Sousa** [2,†], **María Segura-Fornieles** [2] and
**Gemma Vicente** [1,*]

[1]  Department of Chemical, Energy and Mechanical Technology, ESCET, Universidad Rey Juan Carlos,
Móstoles, 28933 Madrid, Spain; irene.megia@urjc.es (I.M.-H.); alejandra.sanchezbayo@urjc.es (A.S.-B.)

[2]  AlgaEnergy S.A. Parque Empresarial La Moraleja, Avda. Europa, 19. Alcobendas, 28108 Madrid, Spain;
fws@algaenergy.es (F.G.W.-S.); msf@algaenergy.es (M.S.-F.)

[3]  Department of Chemical and Environmental Technology, ESCET, Universidad Rey Juan Carlos, Móstoles,
28933 Madrid, Spain; fernando.bautista@urjc.es (L.F.B.); victoria.morales@urjc.es (V.M.)

*  Correspondence: gemma.vicente@urjc.es

†  (F.G.W.-S.) In memoriam.

**Abstract:** *Phaeodactylum tricornutum* is an interesting source of biomass to produce biocrude by hydrothermal liquefaction (HTL). Its biochemical composition, along with its biomass productivity, can be modulated according to this specific application by varying the photoperiod, the addition of $CO_2$ or the variation of the initial nitrate concentration. The lab-scale culture allowed the production of a *P. tricornutum* biomass with high biomass and lipid productivities using a 18:6 h light:dark photoperiod and a specific $CO_2$ injection. An initial concentration of nitrates (11.8 mM) in the culture was also essential for the growth of this species at the lab scale. The biomass generated in the scale-up photoreactor had acceptable biomass and lipid productivities, although the values were higher in the biomass cultivated at the lab scale because of the difficulty for the light to reach all cells, making the cells unable to develop and hindering their growth. The biocrudes from a 90 L cultivated microalga (B-90L) showed lower yields than the ones obtained from the biomass cultivated at the lab scale (B-1L) because of the lower lipid and high ash contents in this biomass. However, the culture scaling-up did not affect significantly the heteroatom concentrations in the biocrudes. A larger-scale culture is recommended to produce a biocrude to be used as biofuel after a post-hydrotreatment stage.

**Keywords:** culture; scale-up; microalgae; hydrothermal liquefaction; biocrude; *Phaeodactylum tricornutum*

## 1. Introduction

Nowadays, there is an increase in energy demand, which requires the replacement of a high percentage of fossil fuels with green energy supplies and technological development [1]. Thus, many investigations are focused on the search for new ways of producing energy and fuels, the vast majority from renewable sources. In this context, the European Directive on the promotion of the use of energy from renewable sources establishes a binding renewable energy target of at least 32% for the European Union by 2030 and the share of renewable energy within the final consumption of energy in the transport sector being at least 14% by the same year [2].

Microalgae are an interesting alternative source for biofuel production due to their rapid growth rate and their ability to accumulate lipids or carbohydrates [3–7]. Moreover, microalgae have additional advantages: (1) microalga cultivation does not compete with agricultural lands, (2) its use would avoid the conflict with food production, (3) microalgae are capable of growing in various aqueous

media, including sewage or seawater, and (4) they are neutral with respect to $CO_2$ emissions [8–11]. Algae are included as feedstocks for the production of advanced biofuels in the European Directive for the promotion of renewable energy, which comprise a contribution of these advanced biofuels of at least 3.5% as a share of the final consumption of energy in the transport sector in 2030 [2].

The biochemical composition of the microalgae can be adapted for specific purposes through the manipulation of culture conditions [12,13]. Several authors have pointed out that the change in the abiotic factors such as the nutrient composition, temperature, salinity, pH, photoperiod and intensity and quality of light may affect the biochemical composition of the microalgae [9,13,14]. Photoperiod, $CO_2$ injection and the amount of nitrogen available are some of the most important factors on the microalga culture. They have influence on the growth rate and biomass composition, increasing the lipid accumulation or changing the fatty acid profile by nitrogen deficiency in the culture media [15,16]. These factors, in turn, affect the microalgal-derived biofuel production [9,17,18]. The scale-up viability is another key factor on the culture of the microalgae, since large-scale photoreactors are required for the subsequent biofuel production. In this sense, the increase in the bioreactor capacity usually produces several changes in biomass production and composition [19]. For instance, the biomass yield decreases unpredictably when the light-dark interchange does not remain constant and the dark zone increases, which occurs when the reactor is scaled up, except when only the bioreactor length increases [20].

Among the different types of microalgae, diatoms are the dominant primary producers in the ocean. These marine species are easily adaptable environmentally and are considered responsible for 25% of global primary productivity of organic compounds [21]. There are many marine diatoms, although one of the most studied is *Phaeodactylum tricornutum* because of its high biomass productivity (235 mg/(L·d)) [22]. Therefore, it can be potentially grown at high quantities in large-scale bioreactors [12]. The typical biochemical profile of this microalgae is 30–70% proteins [23], 10–30% carbohydrates [24] and lipid contents between 20% and 30% [15], being an important source of eicosapentaenoic acid (EPA) and carotenoids [25]. This microalga has a high potential to produce advanced biofuels [25,26].

One of the main problems affecting the biofuels production from microalgae is reducing the economic cost, as 50% of the total required energy for the process is consumed in biomass drying. Hydrothermal liquefaction (HTL) is a process able of converting wet biomass into a biofuel. The biomass conversion technique is carried out in a water medium at temperatures of 200–370 °C and pressures of 10–25 MPa [27]. These conditions are established in order to decompose the biomass and hydrolyze the macromolecules (lipids, proteins and carbohydrates) into smaller organic compounds, which are capable in turn of producing hydrocarbons after decarboxylation and denitrogenation reactions [28]. This process is considered the most promising technique for the conversion of wet microalgal biomass, since the biocrude has a calorific value between 30 and 50 MJ/kg, within the range of that of a petroleum crude but, also, because of the exploitation of the additional phases (aqueous, gas and solid residue) obtained from HTL [29,30]. The aqueous phase has water-soluble products such as alcohols, aldehydes, ketones, carboxylic acids and nutrients. Therefore, this phase can be used in the biogas production or be recirculated to the cultivation phase [31]. The gaseous phase contains a high percentage of $CO_2$, so it can be recirculated to the culture as a supplementary supply of inorganic carbon. Finally, the solid residue usually contains some nutrients so it can be used as fertilizer. Additionally, it can be used as biochar thanks to its high content of carbon [29].

In this research, the culture of the promising marine species *P. tricornutum* was studied for further HTL-processing to produce a high-quality and yield biocrude. Different operating conditions in the culture were studied in a lab-scale bioreactor, such as the photoperiod, injection of $CO_2$ in the media and initial nitrate concentration. In addition, the scale-up of the cultivation stage was performed at the best operating conditions. The effect of scaling up the *P. tricornutum* culture to produce a biocrude by HTL was carried out. This microalga has been little-used for HTL, and no references can be found for the specific objectives of this study [32,33].

## 2. Materials and Methods

### 2.1. Microalga and Chemicals

*Phaeodactylum tricornutum* (CCAP 1055/1) inoculum was supplied by AlgaEnergy S.A. (Alcobendas, Spain). The diatom was grown in Mann and Myers culture medium [34].

The characterization of the biomass was carried out using the following reagents: sulfuric acid, phenol, chloroform, methanol, sodium hydroxide, copper sulphate, sodium carbonate and EDTA sodium salt supplied by Scharlab (Barcelona, Spain) and tryton-X, phenylmethylsulfonyl fluoride, Folin reactant and sodium dodecyl sulfate supplied by Sigma Aldrich (St. Louis, MO, USA). Dichloromethane supplied by Scharlab (Barcelona, Spain) was used in the HTL essays.

### 2.2. Lab-Scale Cultivation Experiments

All the cultivation experiments were carried out in 1 L glass bottles fitted with a bubbling aeration system. In all cases, the initial biomass concentration after inoculation was 0.2 g/L, and the initial sodium nitrate concentration in the medium was 11.8 mM. The illumination system consisted in fluorescent lights yielding 200 $\mu E/(m^2 s)$ at the nearest point on the external surface of the bottles, which was enough for the right growth of the species [35]. The tests were performed in triplicate at a constant temperature of 23 °C, and the cultures were run for 14 days. For the study of the effect of photoperiod, two different light:dark values were used, i.e., 12:12 and 18:6 h. The effect of the additional supply of $CO_2$ was also studied for the 18:6 h photoperiod. For this purpose, 1% $CO_2$-enriched air was injected continuously (1 L/min) into the cultures of the bubbling aeration system within the tolerance range [36,37]. The load of nitrate in the media was also assessed in presence of an additional supply of $CO_2$ (1%, 1 L/min) using the following initial nitrate concentrations: 11.8 mM (reference), the minimum value of the optimal range for this species (11.8–17.7 mM) [38], 5.9 mM (50% reduction) and 0 mM (absence of nitrate).

### 2.3. Scale-Up Cultivation Trials

A comparative study of *P. tricornutum* production was carried out by cultivating the microalga during 14 days in a 90 L bubble column photobioreactor to assess the influence of process scaling. The scale-up was carried out under the optimal operating conditions established in the lab-scale cultures (18:6 h lig ht:dark photoperiod using a continuous $CO_2$ flow and an initial nitrate concentration of 11.8 mM). The initial biomass concentration after inoculation was 0.2 g/L, as in the previous lab-scale experiments.

### 2.4. Culture Analysis and Biomass Characterization

Cell growth was measured daily by spectrophotometric absorbance at 625 nm using a spectrophotometer Lange DR 5000 (Hach Lange Spain S.L.U. L'Hospitalet de Llobregat, Barcelona, Spain). Nitrate consumption was monitored every day. For this purpose, a culture sample of 1 mL was centrifuged at 12,000 rpm for 5 min, and then, the absorbance at 220 and 270 nm of the supernatant was measured.

The specific growth rate ($\mu$), defined as the logarithmic increase in cell density per unit of time, was calculated by Equation (1).

$$\mu = \frac{\ln(N_t/N_0)}{\Delta t} \tag{1}$$

where $N_t$ is the biomass concentration at the end of the exponential phase, $N_0$ is the biomass concentration at the beginning of the exponential phase and $\Delta t$ is the time interval ($t$–$t_0$).

For biochemical analysis, dry biomass cells (0.5 g) were lysed with 5 mL of lysis buffer (distilled water containing 1.1 mM EDTA disodium salt, 0.2 mM phenylmethylsulfonyl fluoride and 0.5% Triton-X100). Proteins were measured by the Lowry method [39], adding 100 $\mu$L of sodium dodecyl sulphate solution and 1 mL of Lowry reactive to the lysis supernatant. After 10 min in

darkness, Folin reactive was added and kept for 30 min. Protein concentration was calculated, reading at 750 nm and using bovine serum albumin as a standard calibration protein. Carbohydrates were measured by the Du Bois method [40] using 0.2 mL of the lysed sample, 50 μL of phenol (90%) and 5 mL of sulfuric acid (98%). After 30 min at room temperature, the carbohydrate content was measured by spectrophotometric analysis at 485 nm using glucose as the calibration standard. Lipids were extracted using a chloroform:methanol (4/5 *v/v*) solvent mixture, and the lipid content was calculated gravimetrically [41]. The ash content was determined by gravimetric analysis after calcination at 600 °C for 4 h with a ramp of 50 °C per minute.

The elemental analysis of the biomass was performed in a Flash 2000 (Thermo Fisher Scientific, Waltham, MA, USA) fitted with a thermal conductivity detector (TCD). The C, H, N and S contents were determined by an oxidation/reduction reactor at 900 °C, while the O content was independently determined through a specific pyrolysis reactor at 1060 °C.

The moisture content was studied by drying 0.5 g samples in an oven at 90 °C during 48 h until constant weight.

The productivities of the biomass ($P_B$) and lipids ($P_L$) contained in the biomass were evaluated from Equations (2) and (3), respectively [42].

$$P_B = \frac{N_t - N_0}{\Delta t} \tag{2}$$

where $N_t$ is the biomass concentration at the end of the exponential phase, $N_0$ is the initial biomass concentration (g/L) and $\Delta t$ is the time interval ($t_t$–$t_i$).

$$P_L = P_B \times \frac{\text{mass of lipids}}{\text{mass of dry microalgae}} \tag{3}$$

## 2.5. Hydrothermal Liquefaction Process

The HTL reactions were performed in a 100 mL stainless steel autoclave (EZ-SEAL®, Autoclave Engineers, Erie, PA, USA) using a *P. tricornutum* biomass from the cultivation using 1 L and 90 L photobioreactors. The reactions were carried out at a temperature of 320 °C with 30 g of wet biomass, whose dry weight was 7.74%, at three reaction times (0, 10 and 30 min) in order to evaluate the influence of the reaction time on the quality and quantity of biocrude produced. In this work, time 0 was considered when the reactor reached the temperature setpoint.

At the end of the reaction, the autoclave was rapidly cooled down to room temperature to quench the reaction. Then, the gas phase was allowed to flow through a valve that connected the reactor to the gas chromatograph. Dichloromethane was used to collect the reaction mixture that consisted of solid residue, an aqueous phase and biocrude. The solid residue was separated by vacuum filtration, dried and weighted. The biocrude and aqueous phase were separated by decantation in a separation funnel, and finally, water and dichloromethane were evaporated from their respective phases in order to measure the dry weight of the aqueous phase and biocrude, respectively.

The yields of biocrude ($Y_B$), water-soluble products ($Y_{AP}$) and solid residue ($Y_{SR}$) were determined on a dry basis using Equations (4) to (6) and the overall liquefied phase through Equation (7) [43]. The yield of the gas phase was estimated by difference to the initial dry biomass used.

$$Y_B(\%) = \frac{\text{mass of biocrude}}{\text{mass dry microalgae}} \times 100 \tag{4}$$

$$Y_{AP}(\%) = \frac{\text{mass of aqueous phase}}{\text{mass dry microalgae}} \times 100 \tag{5}$$

$$Y_{SR}(\%) = \frac{\text{mass of solid residue}}{\text{mass dry microalgae}} \times 100 \tag{6}$$

$$\text{Liquefied phase } (\%) = (1 - \frac{\text{mass of solid residue}}{\text{mass dry microalgae}}) \times 100 \tag{7}$$

### 2.6. Analysis of HTL Products

The biocrude was analyzed by elemental analysis following the same protocol as that described above for biomass (Section 2.4). The high heating value (HHV) for the biocrude and the microalgal biomass was determined by using Equation (8) [44].

$$\text{HHV(MJ/kg)} = 0.3414 \times C + 1.4445 \times H - \frac{N - O - 1}{8} + 0.093 \times S \tag{8}$$

Finally, the energy recovery (ER) was calculated with Equation (9).

$$\text{ER } (\%) = \frac{(\text{HHV of biocrude} \times \text{mass biocrude})}{(\text{HHV of microalgae} \times \text{mass of dry microalgae})} \times 100 \tag{9}$$

The total organic content (TOC) of the aqueous phase was determined in Shimadzu-V equipment (Shimadzu Corp, Kyoto, Japan), and the pH was measured with a Basic 30 pH meter (Crison Instruments, Barcelona, Spain).

The gas products were analyzed using a gas chromatograph Varian CP-4900 MicroGC (Varian Inc., Palo Alto, CA, USA) fitted with a thermal conductivity detector (TCD) connected online to the autoclave reactor.

### 2.7. Statistical Analysis

All the experiments were performed in triplicate in order to determine the variability of the results and to assess the experimental errors. In this way, the arithmetical averages and the standard deviations were calculated for all the results.

In addition, statistical analysis was performed by one-way ANOVA using Statgraphics Centurion XVII software (Statpoint Technologies Inc., Warrenton, VA, USA) to determine differences in the biochemical and elemental compositions, yields and productivities between different photoperiods, $CO_2$ injections, nitrate concentrations and cultivation scales. Previously, the variances were checked for homogeneity by the Levene's test, and the Student-Newman-Keuls (SNK) test was used to discriminate among different treatments after a significant F-test. The Student-Newman-Keuls test was used to discriminate among different treatments. Except where another value was explicitly indicated, the confidence level was set at 95% (*p*-value < 0.05).

## 3. Results and Discussion

### 3.1. Lab-Scale Culture

#### 3.1.1. Effect of Photoperiod

The growth of the microalgae under photoperiods 12:12 and 18:6 h of light:dark in the 1 L bioreactor is shown in Figure 1. Maximum biomass concentrations of 1.56 and 1.73 g/L were obtained for the 12:12 and 18:6 h light:dark photoperiods, respectively. Both biomass concentrations were higher than those obtained by Morais et al. [45] with the same microalga, who obtained a maximum concentration of 1.3 g/L under 12:12 and 24:0 h light:dark photoperiods at a lower light intensity of 74 μE/(m$^2$s). However, the volume of the photobioreactor was somewhat higher (2 L) to the one studied in this work. The aeration homogenizes the culture more efficiently in the smaller photobioreactors, which means that all the cells remain for less time in dark areas.

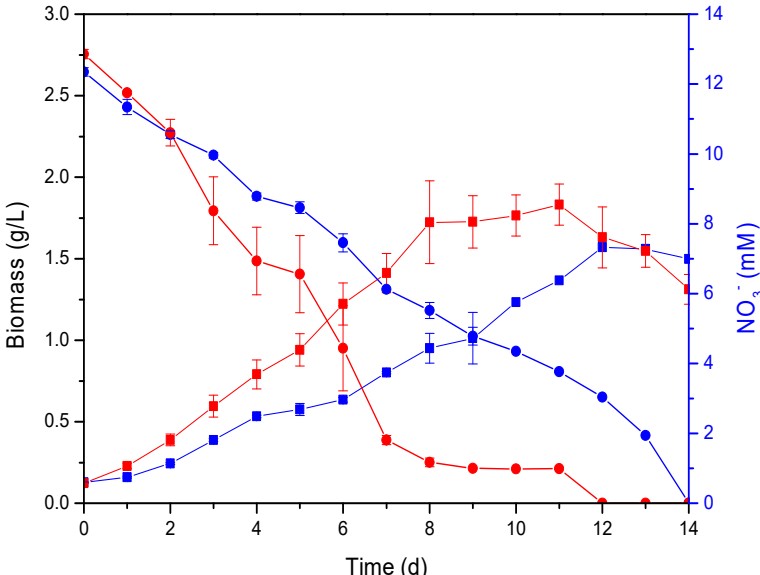

**Figure 1.** Growth curves (squares) and nitrate uptakes (circles) for cultures of *Phaeodactylum tricornutum* at different light:dark photoperiods: 12:12 h (blue symbols) and 18:6 h (red symbols).

The growth rate was 0.24 and 0.33 d$^{-1}$ (*p*-value = 0.01) for the 12:12 and 18:6 h light:dark photoperiods, respectively. Therefore, the growth rate raised with the light hours, which has been previously reported. Thus, a similar increase with the light cycle was also observed for *Chlorella vulgaris* [9,46] and *Nannochloropsis* [47]. In addition, a growth rate of 0.34 d$^{-1}$ for *Nannochloropsis* at the 18:6 photoperiod was previously reported [47]. The most notable difference was observed in the day the stationary phase was reached: 8 and 12 days for the 18:6 and 12:12 h photoperiods, respectively. In both photoperiods, the total nitrate consumption was reached after 14 days of cultivation; however, the number of light hours accelerated the consumption of nutrients.

The lipids obtained increased (significantly at a confidence level 0 94%, *p*-value = 0.055) from 24% ± 1% to 30% ± 3% (Table 1) as the light exposure increased from 12 to 18 h, showing a similar trend than that reported for other species in previous studies [48]. Thus, lipids were mainly accumulated after light exposures above 12 h. In addition to the light time, the amount of lipids depends also on other factors like the $CO_2$ supply or concentration of nutrients in the culture [49]. The amount of lipids in the biomass obtained in the culture is frequently analyzed in the literature. However, a complete study of the biomass composition and productivity, which has been scarcely reported previously, is recommended to fully understand their influence in the biocrude production by HTL. Consequently, the concentration of proteins, carbohydrates and ash in the *P. tricornutum* biomass were measured and presented also in Table 1, together with the corresponding biomass and lipid productivities. Unlike what was observed for *Scenedesmus obliquus* and *Chlorella* [50,51], the protein content did not change significantly (*p*-value = 0.127) when the photoperiod increased the light times. By contrast, the content of carbohydrates decreased (*p*-value = 0.019) as the light hours increased. George et al. [52] obtained similar results in cultures of *Ankistrodesmus falcatus*. Biller and Ross [53] concluded that there is a trend in the biocrude yield from HTL where lipid contents are the main positive influential factor, followed by protein and carbohydrate contents in this order of influence. Thus, a high content of lipids and proteins favors the production of hydrocarbons after hydrolysis and denitrogenation reactions and hydrolysis and decarboxylation, respectively [28].

**Table 1.** Effect of the photoperiod on biomass composition and productivities. $P_B$: biomass and $P_L$: lipid productivities.

| | Light:Dark Photoperiod (h) | |
|---|---|---|
| | **12:12** | **18:6** |
| Lipids (%) | $23.96 \pm 1.09$ | $29.55 \pm 2.73$ |
| Proteins (%) | $49.10 \pm 0.79$ | $54.45 \pm 3.76$ |
| Carbohydrates (%) | $6.66 \pm 0.80$ | $3.73 \pm 1.02$ |
| Ash (%) | $20.27 \pm 0.56$ | $12.27 \pm 0.04$ |
| $P_B$ (mg/(L·d)) | $123.87 \pm 0.06$ | $200.02 \pm 15.18$ |
| $P_L$ (mg/(L·d)) | $25.06 \pm 0.01$ | $52.67 \pm 4.00$ |

The ash content decreased from $20.27\% \pm 0.56\%$ to $12.27\% \pm 0.04\%$ ($p$-value = 0.002) with the increase in light exposure. The ash portion varies according to the culture conditions [54], and in this case, the observed reduction of ash was probably due to the corresponding increase in the photosynthetic efficiency, which, in turn, means a higher biomass production with the light cycle. A high ash content may inhibit the transformation of the microalgae in HTL and has a negative effect on the biocrude properties [55].

Interestingly, the biomass productivity increased at higher light photoperiods, which was observed previously for the microalga *S. obliquus* [50]. The biomass productivity achieved $123.87 \pm 0.06$ and $200.02 \pm 15.18$ mg/(L·d) ($p$-value = 0.013) at the 12:12 and 18:6 h light:dark photoperiods, respectively. As the lipid content and biomass productivity raised with the light, the lipid productivity also increased with the hours of light ($25.06 \pm 0.01$ to $52.67 \pm 4.00$ mg/(L·d) ($p$-value < 0.007) for the 12:12 and 18:6 h photoperiods, respectively). Therefore, the results showed a greater efficiency in the production of both biomass and lipids when the culture was exposed to 18 h of light.

One of the objectives of the liquefaction process is to obtain a biofuel with a low content of heteroatoms, mainly O and N. Table 2 summarizes the biomass elemental analysis at the different photoperiods. The O and N contents of the biomass obtained at both photoperiods did not show significant differences, which was related with the biochemical composition of the biomass obtained. The contents of these heteroatoms were lower than the reported values of other microalgae, such as *Arthrospira platensis* [56], *C. vulgaris* [53] and *Dunaliella tertiolecta* [33].

**Table 2.** Effect of the photoperiod on the elemental composition of the biomass.

| Light:Dark (h) | C (%) | N (%) | H (%) | S (%) | O (%) |
|---|---|---|---|---|---|
| 12:12 | $53.65 \pm 0.80$ | $9.29 \pm 0.10$ | $7.65 \pm 0.10$ | $0.12 \pm 0.01$ | $29.29 \pm 0.60$ |
| 18:6 | $52.74 \pm 0.20$ | $9.18 \pm 0.10$ | $7.50 \pm 0.10$ | $0.90 \pm 0.00$ | $29.68 \pm 0.30$ |

Therefore, biomass grown under a larger number of light hours seems a priori more adequate to HTL due to its high biomass and lipid productivities, high protein and lipid contents and lower ash content. Thus, the 18:6 h photoperiod was chosen to produce the *P. tricornutum* biomass for HTL.

### 3.1.2. Influence of $CO_2$ Injection

The growth curve (Figure 2) shows the results obtained in the essays with and without an additional $CO_2$ injection in the *P. tricornutum* culture. It can be observed that the exponential phase was extended for two more days when a continuous flow of additional $CO_2$ was injected. This caused a remarkable increase in the total biomass production, reaching a concentration above 2.61 g/L with a continuous injection of $CO_2$ compared to 1.73 g/L obtained in the absence of the supplemental $CO_2$. A similar notable increase was observed previously for the other diatom [17]. Total nitrate

depletion was reached at the end of both experiments (Figure 2), as observed in the study of the effect of the photoperiod.

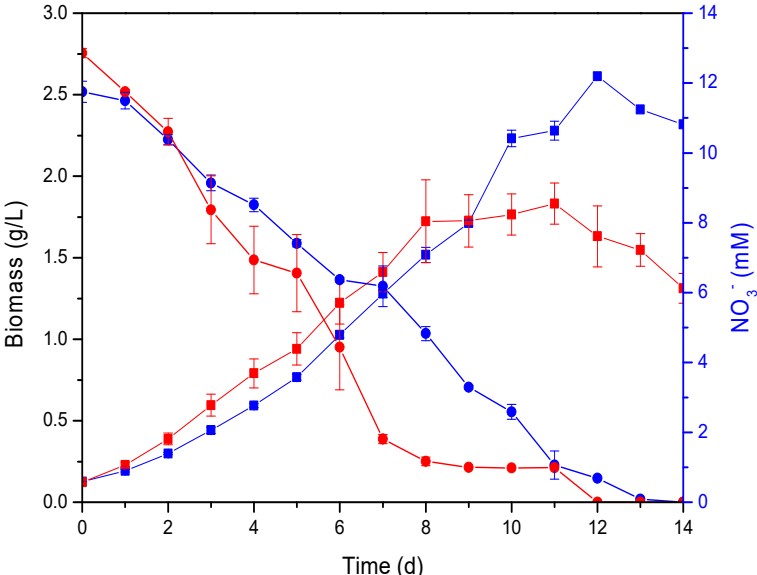

**Figure 2.** Growth curves (squares) and nitrate uptake (circles) for cultures of *P. tricornutum* under an 18:6 h photoperiod with continuous $CO_2$ injection (blue symbols) and without $CO_2$ injection (red symbols).

Table 3 shows the biochemical composition and biomass and lipid productivities of *P. tricornutum* microalgae grown with and without supplemental $CO_2$ injections. The results show that the contents of lipids and proteins did not change significantly (*p*-values: 0.171 and 0.121, respectively) when a flow of pure $CO_2$ was bubbled in the culture. This could be due to the fact that the metabolic flow involving the biosynthesis of these types of biomolecules is not altered if the carbon source is maintained above a minimum threshold. Other authors, however, reported an increase in lipid and protein accumulations when $CO_2$ was injected in the culture medium up to a maximum concentration of 10% [17]. Carbohydrates increased with the injection of $CO_2$ into the culture medium, from 3.73% ± 1.02% to 12.54% ± 0.68% (*p*-value = 0.0005). This is in accordance with a previous study for the microalga *S. obliquus* [57] where the availability of $CO_2$ in the culture medium favored the production of carbohydrates during the dark phase.

**Table 3.** Effect of $CO_2$ injection on biomass composition and productivities.

|  | Without $CO_2$ Injection | With $CO_2$ Injection |
|---|---|---|
| Lipids (%) | 29.55 ± 2.73 | 33.15 ± 2.56 |
| Proteins (%) | 54.45 ± 3.76 | 49.16 ± 1.59 |
| Carbohydrates (%) | 3.73 ± 1.02 | 12.54 ± 0.68 |
| Ash (%) | 12.27 ± 0.04 | 5.15 ± 1.46 |
| $P_B$ (mg/(L·d)) | 200.02 ± 15.18 | 210.54 ± 6.12 |
| $P_L$ (mg/(L·d)) | 52.67 ± 4.00 | 69.80 ± 1.68 |

The ash concentration decreased with the $CO_2$ supplement from 12.27% ± 0.04% to 5.15% ± 1.46% (*p*-value = 0.014). The acid nature of $CO_2$ in the solution controls the pH of the medium at lower values with the supplementation of carbon dioxide, avoiding the precipitation of insoluble salts of the medium during the growth [58].

However, the productivity of the biomass remained without significant changes (*p*-value = 0.353), regardless of whether the additional $CO_2$ was injected or not. Although an inhibition by an excess of $CO_2$ to the culture medium was previously reported [59], the results of the biomass productivity did not indicate an inhibition effect by the presence of the extra $CO_2$. These similar values of biomass productivities may be due to the fact that the microalgae grew faster and took two more days to reach the stationary phase in the presence of extra amounts of $CO_2$.

The $CO_2$ supplemented to the microalgae system led to a significant rise in lipid productivity, from 52.67 ± 4.00 to 69.80 ± 1.68 mg/(L·d) (*p*-value = 0.010), despite the similar biomass productivities obtained. A similar increase with $CO_2$ was observed for *C. vulgaris* [60].

The elemental composition of the microalgal biomass (Table 4) slightly changed when the $CO_2$ availability increased. The results only showed a small increase in the O content (*p*-value = 0.033) and a small decrease in the N content (*p*-value = 0.041). N is mainly contained in proteins, which remained approximately constant with the additional $CO_2$ injection. Therefore, the above observed results were likely due to the remarkable increase in carbohydrates, rich in O, in the biomass cultured with extra $CO_2$. Therefore, there is no significant modifications regardless of the presence of a specific $CO_2$ injection in the culture medium, because the carbon content from air allows a nonlimiting growth of cells [61]. Similar results have been reported for *C. vulgaris* [59].

**Table 4.** Effect of $CO_2$ injections on the elemental composition of the biomass.

|  | C (%) | N (%) | H (%) | S (%) | O (%) |
|---|---|---|---|---|---|
| Without $CO_2$ injection | 52.74 ± 0.20 | 9.18 ± 0.10 | 7.50 ± 0.10 | 0.90 ± 0.00 | 29.68 ± 0.30 |
| With $CO_2$ injection | 53.13 ± 0.06 | 8.52± 0.02 | 7.33 ± 0.01 | 0.98 ± 0.01 | 30.10 ± 0.05 |

Therefore, an extra $CO_2$ injection was selected to obtain the *P. tricornutum* biomass for the subsequent larger-scale culture and biocrude production by HTL because of the high lipid productivity, together with a lower ash content achieved in the experiments with a direct supplement of $CO_2$.

### 3.1.3. Effect of Initial Nitrate Concentration

Based on the previous cultures with 18:6 h of a light:dark photoperiod and $CO_2$ injection, the initial nitrate content was reduced with respect to the original culture medium (11.8 mM) to stress the microalga and evaluate its growth, composition and productivity. The results show that the decrease in nitrate negatively affected the biomass concentration (Figure 3), since the cell growth was inhibited by the lack of a readily available N source. Similar results were observed earlier [62,63]. Culture mediums with nitrate reductions of 50% implied a 40% decrease in biomass growth (from 2.61 g/L to 1.56 g/L). In the culture media without nitrates, the biomass growth decreased to 0.86 g/L. Our results indicate that nitrates can be considered as essential nutrients for *P. tricornutum* growth [62], and therefore, there is a need to have an initial nitrate concentration of at least 11.8 mM in the media.

The original medium (nitrate concentration of 11.8 mM) reached the stationary phase on day 10, whereas the total nitrates uptake was reached on day 13. On the other hand, the culture with a 50% of nitrates reduction reached the stationary phase on day 9 because of the nitrogen limitation. The concentration of nitrates was exhausted on day 8, and therefore, cells could not reproduce. In the medium without them, the stationary phase was reached on day 6 because of the inhibitory effect of the lack of a nitrogen source.

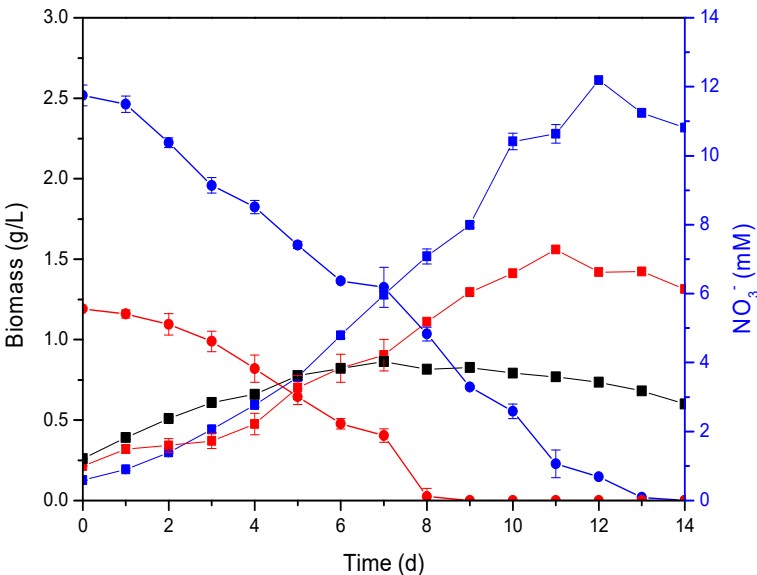

**Figure 3.** Growth curve (squares) and nitrate uptake (circles) for cultures of *P. tricornutum* under a 18:6 h photoperiod with continuous $CO_2$ injection starting with different initial nitrate concentrations: 11.8 mM (blue symbols), 5.9 mM (red symbols) and 0 mM (black symbols).

Table 5 shows the effect of the initial nitrate concentration in the culture in the biomass composition. The final lipid content was not significantly affected by the reduction of the initial concentration of the nitrogen source. Although microalgae are subjected to stress in the absence of nitrates, favoring the accumulation of lipids in the cells [64,65], the harvesting and characterization of the biomass was carried out after 14 days of cultivation. At that time, nitrate was depleted from the media in all the experiments, so that all cultures were similarly stressed in terms of N availability. The concentration of proteins decreased from 49.16% ± 1.59% (nitrate concentration of 11.8 mM) to 33.47% ± 2.38% (lack of nitrate). The presence of nitrate in the medium is the main source of N assimilation for the microalgae and, therefore, essential for protein formation [62].

**Table 5.** Effect of the initial nitrate concentration on the biomass composition and productivities.

|  | Initial ($NO_3^-$) (mM) | | |
|---|---|---|---|
|  | **11.8** | **5.9** | **0** |
| Lipids (%) | 33.15 ± 2.56 a | 34.89 ± 2.25 a | 34.91 ± 2.01 a |
| Proteins (%) | 49.16 ± 1.59 a | 43.81 ± 2.68 b | 33.47 ± 2.38 c |
| Carbohydrates (%) | 12.54 ± 0.68 a | 11.96 ± 0.01 a | 22.65 ± 2.12 b |
| Ash (%) | 5.15 ± 1.46 a | 9.34 ± 0.78 b | 8.97 ± 0.62 b |
| $P_B$ (mg/(L·d)) | 210.54 ± 5.08 a | 119.75 ± 0.24 b | 84.21 ± 2.18 c |
| $P_L$ (mg/(L·d)) | 69.80 ± 1.68 a | 41.78 ± 0.08 b | 29.40 ± 0.76 c |

Values show average ± standard deviation, and letters show significant differences between different initial nitrate concentrations for each component and productivity (*p*-value < 0.05, Student-Newman-Keuls (SNK) test).

The amount of carbohydrates was significantly higher (22.65% ± 2.12%) for the cultivation performed in the absence of an initial supply of nitrate than that obtained when the initial concentrations of nitrate were 11.8 and 9.6 mM (12.54% ± 0.68% and 11.96% ± 0.01%, respectively). Nutrient reductions such as nitrate drive the microalgae to accumulate energy-rich reserve compounds—essentially, lipids and carbohydrates [64].

The ash concentration increased with the reduction of nitrate in the culture from 5.15% ± 1.46% (11.8 mM) to 9.3%5 ± 0.78% (5.9 mM) and 8.97% ± 0.62% (absence of an initial supply of nitrate).

Therefore, the reduction of the N source negatively affected the biomass composition, increasing the ash content, which is not adequate for the following HTL stage. The lower ash content obtained with the higher initial nitrate concentration (11.8 mM) was due to the better photosynthetic efficiency and, therefore, higher biomass production at this nitrate concentration.

Regarding the biomass and lipid productivity, Table 5 shows about a 2.5-fold increase in these values at the higher initial nitrate concentration. Although the composition of the biomass is affected at different levels, the main effect of the availability of large amounts of N was exerted on the growth of microalgae, boosting the values of the specific growth rate and, hence, the productivity associated to the biochemical components of the cells.

From the results obtained in the elemental analysis of the biomass (Table 6), the main remarkable effect of the decrease in the amount of N in the growth medium was a significant reduction in the N content of the biomass when the culture was not supplemented with an initial nitrate concentration. The rest of the elements underwent changes that, although statistically significant in some cases, did not turn out to be very noteworthy.

**Table 6.** Effect of the initial nitrate concentration on the elemental composition of the biomass.

| $(NO_3^-)$ (mM) | C (%) | N (%) | H (%) | S (%) | O (%) |
|---|---|---|---|---|---|
| 11.8 | 53.13 ± 0.06 a | 8.52± 0.02 a | 7.33 ± 0.01 a | 0.98 ± 0.01 a | 30.10 ± 0.05 a |
| 5.9 | 53.58 ± 0.20 b | 9.27 ± 0.03 b | 7.38 ± 0.19 a | 1.53 ± 0.17 b | 28.23 ± 0.10 b |
| 0 | 59.07 ± 0.15 c | 2.77 ± 0.01 c | 8.81 ± 0.03 b | 0.42 ± 0.03 c | 28.93 ± 0.15 c |

Values show average ± standard deviation, and letters show significant differences between different initial nitrate concentrations for each element ($p$-value < 0.05, SNK test).

According to the drastic reduction in biomass and lipid productivities and the increase of ash content in the absence of nitrates, the importance of the nitrate presence was confirmed for the adequate *P. tricornutum* growth. In this sense, an initial concentration of 11.8 mM of nitrate in the growth medium was chosen to continue the study.

### 3.2. Culture Scaling

Scaling tests (Figure 4) show that the microalga did not require an adaptation phase, regardless of the reactor used, and began to grow rapidly in the culture medium. The stationary phase was reached four days later in the 90 L volume reactor compared to in the 1 L one. In addition, the specific growth rate was 0.29 and 0.17 $d^{-1}$ in the bioreactors of 1 L and 90 L, respectively, which are suitable values for this type and size of bubble column reactors indoors [66]. The biomass produced in the 90 L bioreactor is close to other productions obtained outdoors in similar column bioreactors for this species (0.96 g/L) [25], as well as for other pilot plant reactors, such as circular ponds or tubular photobioreactors [21], indicating a good biomass production. It must be noted that total consumption of nitrate was not reached for the 90 L culture after 14 days.

The value of the biomass productivity obviously decreased with the bioreactor volume because of the observed greater shielding of the microalga, which prevents the light from reaching the microalga cells at higher volumes (Table 7). Thus, the biomass productivities were 210.54 ± 5.08 and 56.01 ± 4.45 mg/L·d for the 1 L and 90 L bioreactors, respectively ($p$-value < 0.0001). In addition, a light stress reduction at the higher volume, produced by the increase in dark areas, affected the lipid productivity, decreasing from a value of 69.8 ± 1.68 for the 1 L reactor to 9.85 ± 0.78 mg/(L·d) for the 90 L ($p$-value < 0.0001). The high biomass and lipid productivities obtained at the 1 L bioreactor can be achieved using optimal culture conditions, which only have a remarkable positive effect at this small-scale cultivation. In this sense, the productivities obtained on a larger scale, although lower, results are adequate for the production of this microalgae and its subsequent use in the production of biofuels.

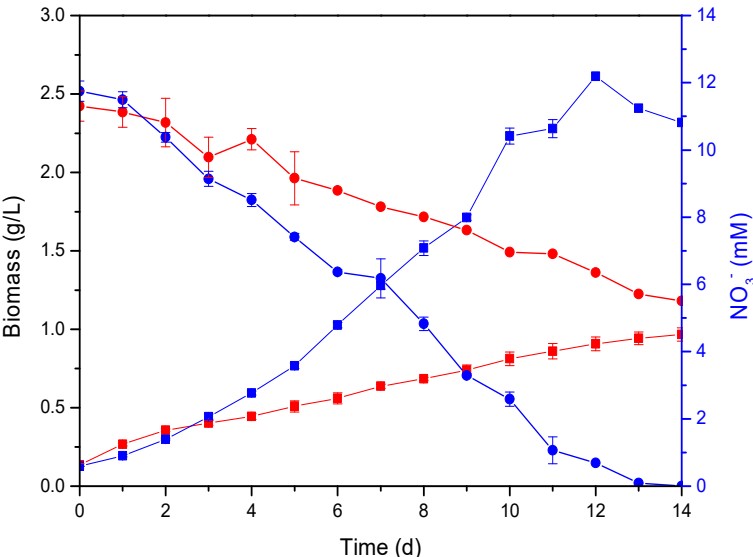

**Figure 4.** Growth curves (squares) and nitrogen uptake (circles) for cultures of *P. tricornutum* under an 18:6 h photoperiod with continuous $CO_2$ injections in photobioreactors at different scales: 1 L bottle (blue symbols) and 90 L column bioreactor (red symbols).

**Table 7.** Biomass analysis for the 1 L bottle culture and the 90 L scale.

| | $V_{reactor}$ (L) | |
|---|---|---|
| | **1** | **90** |
| Lipids (%) | 33.15 ± 2.56 | 17.59 ± 0.03 |
| Proteins (%) | 49.16 ± 1.59 | 58.49 ±0.18 |
| Carbohydrates (%) | 12.54 ± 0.68 | 9.80 ± 0.05 |
| Ash (%) | 5.15 ± 1.46 | 14.12 ± 0.17 |
| $P_B$ (mg/(L·d)) | 210.54 ± 5.08 | 56.01 ± 4.45 |
| $P_L$ (mg/(L·d)) | 69.80 ± 1.68 | 9.85 ± 0.78 |

Finally, the elemental analysis of the biomass produced in both reactors hardly showed significant differences (Table 8). A slight but significant decrease of C and H ($p$-value < 0.0001) was observed in the biomass grown at a photoreactor of 90 L, related to a higher concentration of ashes and a lower concentration of lipids and carbohydrates. In addition, the concentration of N was slightly lower ($p$-value = 0.0001) in the biomass cultivated in the bioreactor of 90 L (7.50% ± 0.02%) in comparison to the biomass obtained in the 1 L reactor (8.52% ± 0.02%) despite the higher concentration of proteins found in the former biomass. Consequently, a little higher ($p$-value = 0.0004) concentration of O was obtained in the biomass grown in a 90 L culture (37.29% ± 0.22%) than that in the corresponding biomass cultivated in the 1 L bioreactor (30.10% ± 0.05%), which is probably connected with the observed increase in proteins as the size of the photobioreactor increased. In this sense, the elemental distribution in the biomass obtained did not change substantially, despite the differences of the biomass analysis obtained (Table 7). Although a small size photobioreactor seems to be more adequate to obtain a biomass for biofuel production, according to the above results of the biomass analysis, the development of the HTL process on an industrial scale requires high amounts of biomass and, therefore, the use of large-scale photoreactors.

**Table 8.** Elemental analysis for the biomass cultivated on different scales.

| $V_{bioreactor}$ (L) | C (%) | N (%) | H (%) | S (%) | O (%) |
|---|---|---|---|---|---|
| 1 | 53.13 ± 0.06 | 8.52± 0.02 | 7.33 ± 0.01 | 0.98 ± 0.01 | 30.10 ± 0.05 |
| 90 | 47.14 ± 0.22 | 7.50 ± 0.02 | 6.94 ± 0.03 | 1.12 ± 0.00 | 37.29 ± 0.22 |

### 3.3. Hydrothermal Liquefaction Process

The yield of the different fractions obtained from the HTL process at 320 °C (biocrude, aqueous phase, gas phase and solid residue) for the three reaction times and both culture scales evaluated are shown in Figure 5. The biomass grown in the 1 L photobioreactor (B-1L) exhibited a higher biocrude yield compared to the biomass produced at 90 L (B-90L) in the HTL process at all reaction times ($p$-value $\leq$ 0.05), which was mainly due to the fact that the former had a higher lipid content (Table 7) that contributed to the increase in the yield of this fraction [53]. The yield of the biocrude fraction obtained from the B-1L at 10 min was similar (36.64% ± 4.93%) to those obtained by Christensen et al. (38.8% ± 1.3% at 350 °C and 15 min) for commercial *P. tricornutum* at harsher operating conditions [32], while those of B-90L were lower (25.61–30.03%).

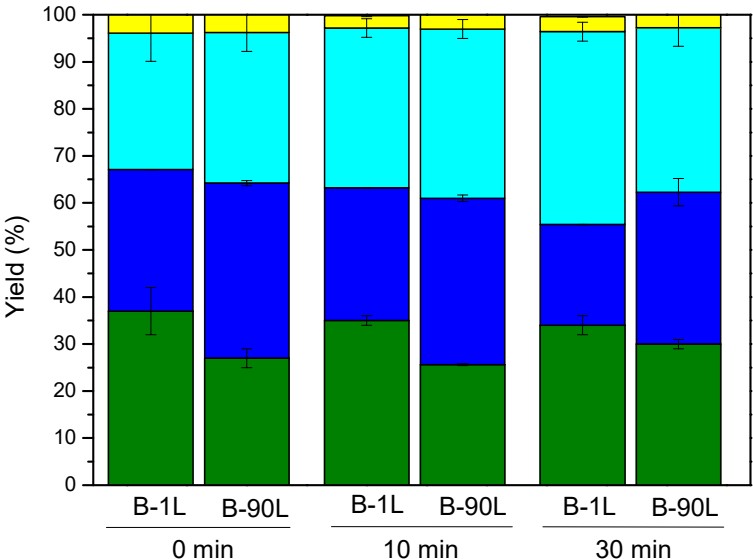

**Figure 5.** Yields of the different phases after a hydrothermal liquefaction (HTL) reaction at 320 °C with the *P. tricornutum* biomass: biocrude (green), aqueous phase (dark blue), gas phase (light blue) and solid residue (yellow). Time 0 min was considered when the reactor reached the setpoint temperature. B-1L and B-90L: *P. tricornutum* biomass cultivated in 1 L and 90 L bioreactors, respectively.

The biocrude yields produced using the B-1L were significantly unaffected at different reaction times (33.99% ± 1.67%–36.64% ± 4.93%, $p$-value > 0.05). Christensen et al. [32] obtained similar yields at 300 and 325 °C, which are closer to the value of 320 °C used in this work. Similarly, the biocrude yield from B-90L did not change with time (25.61% ± 0.27%–30.03% ± 0.95%). Therefore, the reaction time did not have a significant effect on the biocrude yields at the high temperature used in this work (320 °C).

The yield of the aqueous fraction decreased with the reaction time using B-1L but remained approximately constant over time with B-90L. However, the yields of the aqueous fractions were higher ($p$-values < 0.05) for the B-90L because of the lower biocrude yields obtained in this case. More water-insoluble molecules were produced at longer reaction times, which may be due to decarboxylation, deamination, dehydration, oligomerization and condensation reactions, thus producing a change in the product distribution and a decrease in the yield of the aqueous phase [67].

The yield of the gas phase was nearly constant (30.51% ± 0.40% to 35.96% ± 1.98%) for B-90L. A similar behavior were reported previously for HTL carried out at 350 °C and similar reaction times with *Nannochloropsis* [53,68]. However, there was a significant increase in the gas phase with time in the 1 L culture (28.50% ± 5.54% to 41.03% ± 1.77%) related to the higher content of C and, to a lesser extent, of carbohydrates in this biomass, which produces higher gas yields. The production of gaseous compounds is related to the yields of the biocrude and aqueous phases in the reaction mechanism proposed by Sheehan and Savage for *Nannochloropsis* [30].

The solid residues represented the smallest fractions of the products obtained in the HTL process, their yields varying from 2.59% to 4.74%, in agreement with bibliographic results, where solid fraction yields below 10% are usual for the HTL of microalgae [29]. The low yields achieved for the solid residue are the cause for the high transformation efficiency of HTL, showing liquefaction yields (sum of biocrude and aqueous and gas phases) over 95% in all cases [29].

### 3.4. Analysis of Biocrude

Table 9 shows the elemental composition, the higher heating value (HHV) and the energy recovery (ER) in the biocrude obtained by HTL using B-1L and B-90L at different reaction times.

**Table 9.** Elemental analysis (wt%, dry basis), higher heating value (HHV) and energy recovery of the biocrude phase after the hydrothermal liquefaction (HTL) process at 320 °C for different reaction times. B-1L and B-90L: *P. tricornutum* biomass cultivated in 1 L and 90 L bioreactors, respectively. ER: energy recovery.

| Time (min) | Biomass | C (%) | H (%) | N (%) | S (%) | O (%) | HHV (MJ/kg) | ER (%) |
|---|---|---|---|---|---|---|---|---|
| 0 | B-1L | 74.46 ± 1.41 | 9.73 ± 0.47 | 5.60 ± 0.97 | 0.42 ± 0.20 | 9.79 ± 0.20 | 38.27 ± 1.52 | 51.46 ± 1.86 |
| | B-90L | 74.92 ± 0.86 | 9.52 ± 0.09 | 5.43 ± 0.10 | 0.35 ± 0.04 | 9.78 ± 0.10 | 39.36 ± 0.46 | 38.61 ± 0.61 |
| 10 | B-1L | 76.54 ± 0.59 | 9.76 ± 0.07 | 5.52 ± 0.28 | 0.64 ± 0.01 | 7.54 ± 0.01 | 39.47 ± 0.34 | 50.61 ± 0.60 |
| | B-90L | 75.06 ± 0.73 | 9.37 ± 0.09 | 5.39 ± 0.04 | 0.43 ± 0.04 | 9.74 ± 0.12 | 39.71 ± 0.40 | 36.77 ± 0.52 |
| 30 | B-1L | 77.52 ± 0.03 | 9.62 ± 0.12 | 5.43 ± 0.23 | 0.57 ± 0.10 | 6.86 ± 0.31 | 39.37 ± 0.26 | 49.11 ± 0.53 |
| | B-90L | 75.29 ± 0.74 | 9.65 ± 0.07 | 5.42 ± 0.03 | 0.44 ± 0.11 | 9.20 ± 0.06 | 38.767 ± 0.37 | 41.98 ± 0.50 |

A significant decrease in the O amount was observed in all the biocrude phases with respect to the starting biomass. The O content varied with time from 9.79% ± 0.20% to 6.86% ± 0.31% and from 9.78% ± 0.10% to 9.20% ± 0.06% in the biocrudes obtained in the HTL of the B-1L and B-90L, respectively, whereas this heteroatom content was 30.10% ± 0.05% and 37.29% ± 0.22% in the starting biomass obtained in the same bioreactors. The O concentration was moderately higher for the biocrudes from B-90L, particularly at longer HTL times. These results indicated the presence of decarboxylation reactions during the HTL process that intensified while increasing the reaction time [69]. Furthermore, these values were lower than the O contents in the biocrude obtained in the HTL with other species, such as *Tetraselmis* (12.3%), *Scenedesmus almeriensis* (9.6%), *Chlorella* (30.38%), *Nannochloropsis gaditana* (14.49%) [27] or *Scenedesmus* (10.5%) [70].

The N content in the biocrudes were very similar in all cases and lower (5.42% ± 0.03% to 5.39% ± 0.04%) than the corresponding N amount in the two initial biomasses (8.52% ± 0.01% and 7.50% ± 0.02% for the biomasses obtained in the reactors with 1 and 90 L, respectively) because of the denitrogenation reactions during HTL [27]. These values were comparable to those obtained for the same species at 325 °C (5.62%) [32].

The observed decrease in N and O amounts in the biocrudes is typical of the HTL process of microalgae, which causes the contents of C (74.46% ± 1.41%–77.52% ± 0.03%) and H (9.37% ± 0.09%–9.76% ± 0.07%) to increase with respect to the corresponding C and H amounts in the raw biomass. The concentrations of C and H changed within the range 53.13% ± 0.06%–47.14% ± 0.22% and 7.33% ± 0.01%–6.94% ± 0.03% for the 1 and 90 L cultivated microalgae, respectively. As noted above, the composition of the initial biomass hardly interfered with the C and H contents in the biocrudes, since these values were very similar. Interestingly, the amounts of C and H of the biocrudes described herein were considerably higher than those recently reported for the same microalga [33], where the biocrudes with slightly lower N percentages and appreciably higher concentrations of O were obtained. However, the elemental composition indicates that the biocrudes cannot be directly used as transport fuel, and, consequently, a subsequent hydrotreating stage would be required to reduce the contents of N and O and, therefore, improve the chemical composition of the biocrudes to fulfil the standard regulations concerning the presence of these heteroatoms in the commercial fuel.

The calculated HHV values of the biocrudes for each of the HTL reactions of both biomasses were similar (38.27 ± 1.52–39.71 ± 0.50 MJ/kg), within the range obtained from other microalgae (30–43 MJ/kg) [28] and close to petroleum crude oil (43 MJ/kg) [71]. In addition, the values of HHV were significantly higher with respect to the corresponding values in the initial microalgal biomass (27.25–26.81 MJ/kg), due to the decrease in the O content and the increase of C and H contents in the biocrudes. The calorific value obtained were higher than that found by López-Barreiro et al. (30.3 and 35.9 MJ/kg) for biocrude from HTL produced at similar temperatures with *P. tricornutum* cultured in bubble columns of 25 L [33]. The presence of nutrients from the culture medium in HTL increased the biocrude yields and the contents of C and O in comparison to the HTL of commercial microalga, being responsible for the increase in calorific value [32]. Furthermore, the HHV achieved in the present work were similar to those obtained by Christensen et al. [32] for temperatures around 400 °C.

The energy recovered in the biocrude obtained from the B-90L were lower (38.61–41.98%) than the corresponding values for the B-1L (49.11–51.47%) because of the previously observed lower biocrude yields in the HTL with the B-90L.

The biofuel properties were significantly affected by their H/C, N/C and O/C ratios, as it is well known. The above ratios for the biocrudes, along with those of a petroleum diesel, a biodiesel and a biocrude obtained from the microalgae *N. gaditana* [72], were plotted in Van Krevelen diagrams (Figure 6) for comparison purposes. The results showed that the O/C and N/C ratios decreased in our biocrudes with respect to the initial biomass, which were due to the denitrogenation and decarboxylation reactions taking place during the HTL process [27].

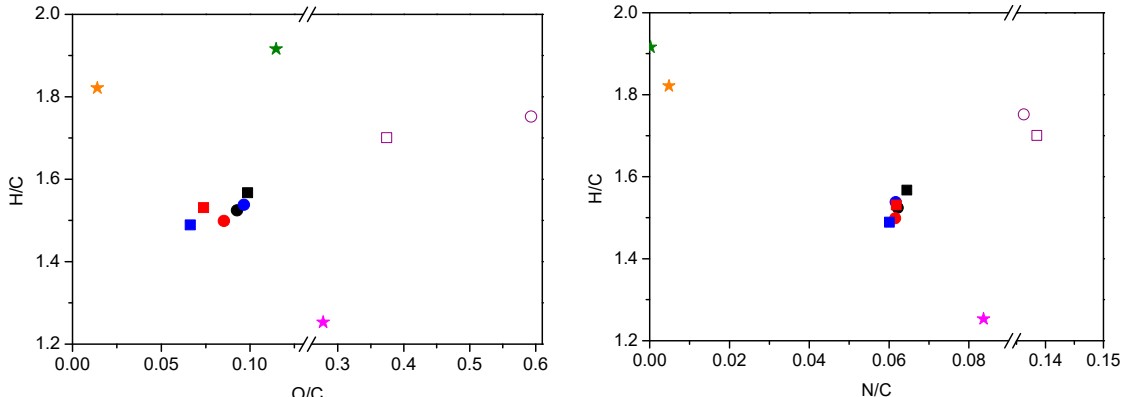

**Figure 6.** Van Krevelen diagrams for biocrude from HTL at 320 °C using B-90L (circles) at different times: 0 (●), 10 (●) and 30 min (●) and B-1L (squares) at different times: 0 (■), 10 (■) and 30 min (■). For (**a**,**b**), values for B-90L (○) and b-1L (□), reference diesel (★), biodiesel from *N. gaditana* oil (★) and biocrude from *N. gaditana* obtained through HTL at 320 °C and 10 min (★) [72]. B-1L and B-90L: *P. tricornutum* biomass cultivated in 1 L and 90 L bioreactors, respectively.

The O/C ratios of the biocrudes decreased by 82.3% and 85.6% with respect to the initial biomass cultivated in the 1 and 90 L photobioreactors, respectively. The O/C ratios obtained (0.066–0.099) (Figure 6a) were, in all cases, within the usual range found in the literature (0.0–0.3) [73], being lower than those obtained for biodiesel (0.11) and *N. gaditana*-derived biocrude (0.28). A decrease in O/C ratio from 0.099 to 0.066 was observed with time in the HTL of the B-1L. Besides, the lowest O/C ratio (0.066) and, therefore, the closest to the reference diesel (0.014) was achieved in the biocrude obtained in the HTL at 30 min using the B-1L.

In the same way, the N/C ratios of the biocrudes (Figure 6b) were very similar to each other (~0.06), regardless of the biomass used. Therefore, the scaling-up did not seem to affect the fuel properties of the biocrudes significantly. A large decrease of 56.6% and 54.8% in the N/C ratios was observed with respect to the raw biomasses obtained in the 1 and 90 L cultivations, respectively. The N/C of the biocrudes decreased slightly with time, being lower than those found for the biocrude produced with

the microalga *N. gaditana* (0.084) but far from the N/C values of the *N. gaditana* biodiesel (0.0004) and the conventional diesel (0.0048) [72]. Furthermore, all the values obtained were between the limits found in the literature (0.056–0.1) [73].

Based on the O/C and N/C ratios of the biocrudes, the biomass of *P. tricornutum* is a promising feedstock for HTL compared to the microalga *N. gaditana*. However, the H/C ratio decreased from 1.70 for the initial microalga to a range between 1.48–1.56 for the biocrudes when using B-1L. A similar decrease occurred with B-90L from 1.75 for the biomass to values around 1.5 for the biocrudes. These decreases were due to the high yields obtained from the aqueous phases. The H/C ratios of the *P. tricornutum*-derived biocrudes were higher than the H/C ratio for a biocrude from *N. gaditana* (1.25) [72]. Therefore, the H/C values of the biocrude from *P. tricornutum* were closer to those of the reference diesel (1.84). All these values were within the range 1.37–1.62 for biocrudes from microalgae with high protein and lipid contents [7].

### 3.5. Analysis of the Aqueous Phase

The aqueous phase is one of the fractions of the HTL process that are part of the liquefied phase [74]. The elemental analysis of these fractions (data not shown) led to a C content between 7.93% $\pm$ 0.15% and 7.99% $\pm$ 0.06% using the B-1L, whereas the C content was slightly higher (8.43% $\pm$ 0.16% to 9.48% $\pm$ 0.66%) with the biomass cultivated in the larger bioreactor. The relatively low amounts of C in the aqueous layer were due to the breakdown of macromolecules into smaller ones that are soluble in water. However the heteroatom (N and O) amounts were relatively high, mainly because of the hydrolysis of carbohydrates and proteins and the subsequent decarboxylation and deamination, which produced N and O compounds soluble in water media [28]. The N content in the aqueous phase decreased with time using both microalgae, due to the fact that the deamination reactions are promoted at longer times [27], but the values were lower in the case of the B-1L (11.57% $\pm$ 0.25% to 8.13% $\pm$ 0.04%) in comparison to those obtained with the B-90L (13.48% $\pm$ 0.66% to 12.56% $\pm$ 0.91%). The opposite trend with time was observed in the O content in the aqueous phase due to the increased decarboxylation of organic molecules [27]. Thus, the O content increased with the HTL time in both cases, showing lower values (29.8% $\pm$ 0.7% to 33.6% $\pm$ 0.5%) when the B-1L was used as compared with B-90L (37.5% $\pm$ 0.4% to 30.7% $\pm$ 0.4%).

The pH values obtained in the aqueous phase were, in all cases, around 8 (7.95–8.77), consistent with the slightly alkaline values found in the literature [29,75], due to the formation of soluble basic compounds in this phase. A slight increase in pH was observed over time for both biomasses, being somewhat higher in the aqueous layer obtained in the HTL of the B-90L (8.33–8.77) than those obtained in the same layer from the B-1L (7.95–8.35). This may be due to the higher amount of N in the aqueous layer in the former, since nitrates were not completely consumed by the microalga cultivated in the 90 L bioreactor. In this sense, the basic composition of the aqueous phase was mainly due to the high portions of $NH_3$ and N compounds [76].

Another key factor in this aqueous phase is the total organic carbon. These values indicated a high formation of new organic compounds soluble in water. The TOC values obtained in this work (880.4–956.2 ppm) were within the bibliographic range (300–1146 ppm) [77].

### 3.6. Analysis of the Gaseous Fraction

Gaseous fractions are also included in the liquefied phase. The gaseous phases obtained by HTL from the different cultivated biomasses had a similar composition. These fractions were mainly composed of $CO_2$ (>80 mol%) in all conditions, as usually reported for the microalgal HTL [74]. The $CO_2$ content of the gaseous fraction was higher than 98 mol% for the HTL reactions using the biomass grown in a 90 L culture. Here, the gas fraction also contained small amounts of linear saturated and monounsaturated light hydrocarbons (C1–C4) (<1 mol%), $H_2$ (<2 mol%) and CO (<1 mol%). In the HTL using B-1L, the $CO_2$ content was lower (79.82–97.19 mol%). In this case, a methane concentration within the range 8–17 mol% was obtained, which may be due to their high content of lipids. Christensen

et al. reported a similar methane content when they performed the HTL at higher temperatures with a biomass with lower concentration of lipids [32]. Additionally, the gas fraction contained small amounts of linear saturated and monounsaturated light hydrocarbons (C2–C4) (<1 mol%), $H_2$ (<2 mol%) and CO (<1 mol%). The large concentration of $CO_2$ of the gas fraction from HTL makes this stream suitable for recirculation towards the cultivation stage in photobioreactors, providing the inorganic carbon source necessary for its development [72].

### 3.7. Analysis of the Solid Fraction

Finally, the smaller phase was the solid residue. This phase consisted mainly of ashes, carbon-rich compounds and minority elements present in the biomass (metals and phosphorous). Therefore, this fraction has been used to obtain biochar from the thermochemical process or in use as a fertilizer [27].

### 4. Conclusions

The composition and productivity of the microalga *P. tricornutum* is affected by the variation in daylight hours, the supply of $CO_2$ into the culture and the availability of nutrients such as nitrate in the culture media. Consequently, it is possible to direct the microalgal metabolism to increase the biomass yield and accumulate specific compounds. For instance, microalgae with high biomass and lipid productivities are adequate for the HTL process to obtain a high yield and quality biocrude. The lab-scale culture of *P. tricornutum* produced a higher biomass and lipid productivities with the daylight hours and a supplemental $CO_2$ injection. In addition, the lab study confirmed that the initial amount of nitrates was essential for efficient growth, achieving high biomass and lipid productivities. The biomass generated during the scaled-up culture had a lower biomass and lipid productivities than the corresponding biomass obtained at the lab scale, despite having carried out the experiments at the same operating conditions. In any case, the values obtained are suitable for the following HTL stage and similar to other works. Using high-volume photobioreactors, a greater shielding of the microalga is frequent and was observed in this work, which prevents the light from reaching the microalga cells, reducing the growth of the microalgae and the lipid accumulation. The biocrudes obtained by HTL using B-90L exhibited somewhat lower yields than those obtained from the biomass cultivated at the lab scale (B-1L), because the former biomass presented lower lipid and higher ash contents. However, the heteroatom contents of the biocrudes were similar and lower than those in the corresponding starting microalga in both cases. Nevertheless, the reduction in the heteroatom content was not enough, and a biocrude post-treatment is required in order to reach the regulated values as direct fuel in transport. In summary, both biocrudes had similar characteristics, although the biomass generated during scale-up had a lower biomass and lipid productivities. In this sense, the use of large-scale cultivated wet biomasses for *P. tricornutum* is recommended to make the overall process more economical and to produce enough amounts of biocrude to be use as a biofuel after the post-hydrotreatment stage.

**Author Contributions:** Conceptualization, F.G.W.-S., G.V. and L.F.B.; methodology, I.M.-H. and A.S.-B.; validation, G.V., V.M., and M.S.-F.; formal analysis, G.V. and V.M.; investigation, I.M.H.; resources, I.M.H. and A.S.-B.; data curation, G.V., L.F.B. and V.M.; writing—original draft preparation, I.M.H. and A.S.-B.; writing—review and editing, G.V., L.F.B. and M.S.-F.; visualization, G.V., V.M. and M.S.-F.; supervision, G.V., V.M. and M.S.-F. and funding acquisition, G.V. All authors have read and agreed to the published version of the manuscript.

**Funding:** The authors acknowledge the support of project IND2017/IND financed by Comunidad de Madrid and the company AlgaEnergy for collaborating in this Industrial Doctorate project. AlgaEnergy acknowledges AENA for the concession of its land to locate the Technological Platform for Experimentation with Microalgae (PTEM).

**Conflicts of Interest:** The authors declare no conflict of interest.

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
