# Peer review of "Scale-Up Cultivation of Phaeodactylum tricornutum to Produce Biocrude by Hydrothermal Liquefaction"

_processes, doi:10.3390/pr8091072_

Round 1

Reviewer 1 Report

The publication of Megia-Hervas et al. investigated optimal cultivation conditions for P. tricornutum in order to increase biomass productivity by varying light cycle and CO2 and nitrate supply. Furthermore, they generated biocrude by hydrothermal liquefaction. The experimental set up and assessed methods seem somehow conclusive. However, some facts such as influence of CO2  etc. on biomass production and composition are not novel. I have the following thoughts and remarks concerning the manuscript

L 62 – 63

The biomass yield is dependent on light dark fluctuations but does not necessarily decrease if the fluctuations do not remain constant. High frequent fluctuations can increase the yield, for instance.  

L 66

Global primary productivity of what?

L 68

Here the areal productivity is given as reference, but in the paper the volumetric productivity is discussed. At some point, it would be easier for the reader if the authors would compare their results in the same units.

L 97 – 99

Please provide composition of the medium. Without knowledge of the used medium and the specific elemental composition especially chapter 3.1.3 can not be retraced. Is there another source of nitrogen?

If the ‘optimized’ medium is not known because of restrictions or whatever the concentrations of the main components could be easily measured via Ion exchange chromatography.

L 109

20 (µmol·m-2·s-1 or µE·m-2·s-1 or µE/(m2s) but not µE/m2s) seems too low and in the range of the light compensation point.
In general, please provide references for the use of cultivation conditions such as nitrate content, CO2 supply etc.

L 119

A scale up from an aerated bottle to a tubular reactor changes more parameters than just the diameter of the tube (mixing, shear stress, global to local aeration, …). These systems are not comparable.

L 125

Typo: microalgae (several times in the text)

L 125 – 128

Growth/dry matter cannot be measured by OD measurements if the intracellular components change over time as the spectral scattering changes with the composition of the algae. Please provide the used wavelengths?

L 191 – 195

Another cultivation is compared without referring to the initial photon flux density. As the growth rate stated in this paper (0.24 and 0.33 d-1) is much lower than other reported growth rates a major light limitation is assumed.

It is physically not possible to homogenize light (lambert beer).

Chapter 3.1.2

CO2 concentrations are not given. kLa values should be measured and carbon transfer rates should be calculated. Without a proper calculation, the results are too qualitative.   

L 308

It is stated that nitrogen is an essential nutrient for P. tricornutum. If you take a closer look at Adenine (C5H5N5), Thymine (C5H6N2O2), Guanine (C5H5N5O) and Cytosine (C4H5N3O) the components of the DNA, it could have been assumed that Nitrogen is essential for all living beings. It makes not sense to cultivate with 0 % nitrate, if there is no additional nitrogen source in the basic medium.
Moreover, if 11,8 mM was detected as minimum (L309 “at least”) in previous studies (no reference provided) why the authors did not apply higher concentrations in order to further increase the productivity?
If there is not nitrogen source, the microalgal cell uses their endogenous amino acids e.g. glucogenic amino acids to produce more important energy sources such as starch (carbohydrates). Would this not be the explanation for the decrease of proteins and increase of carbohydrates (table 5).

L 365

Now the tubular reactor stated in L 119 morphs into a bubble column. It is written that the values for growth of 0.29 d-1 (1 L) and 0.17 d-1 (90 L) are suitable. In the linked references not comparable outdoor PBRs (tubular, open pond) were used resulting in a much lower growth rate (temperature not controlled, light fluctuations) compared to lab scale.

L 373 – 382

The stated optimal culture conditions result in growth rates far lower than the once reported by Geider et al. [1] at very low light levels. As the culture conditions and bioreactors are not described properly the reasons for this low growth rates can only be assumed.

What is meant by light stress reduction? Is there a reference for light stress at 20 µmol·m-2·s-1? Why the authors did not perform continuous illumination?

[1]          R.J. Geider, B.A. Osbonie, J.A. Raven, Growth, Photosynthesis and Maintenance metabolic costs in the diatom Phaeodactylum tricornutum at very low light levels, J Phycol 22 (1986) 39–48. https://doi.org/10.1111/j.1529-8817.1986.tb02513.x.

In general:

  1. Please provide a more detailed description of the methods (especially protein, lipid an carbo determination).
  2. Please use equal number of decimal places for all values.
  3. The sum of all values in the tables is not always 100 % or close to 100 %. Could the authors provide an explanation/suggestion what is the rest of the biomass or is it due to limitations of the applied analytical methods?
  4. Please keep the information about the statistical comparison constant. Sometimes the statistical significance is included in the tables and sometimes not.
  5. Sometimes, literature that is more recent could be used to compare the authors results (e.g. reference 39).

Reviewer 2 Report

The present study by Megía-Hervás et al. described the Scale-up cultivation of Phaeodactylum tricornutum to produce biocrude by hydrothermal liquefaction.

Although the authors claim the novelty of their research, several studies are already available in literature on the same microalga species cultured under similar cultivation variables (Fernández et al. 2000; Acien et al. 2003; Benavides et al. 2013; Gao et al. 2017; Branco-Vieira et al. 2018; Quelhas et al. 2019) and for similar purposes (Christensen et al. 2014).

Furthermore, the novelty of this paper is not much elevated when compared to previous manuscripts.

Please, could authors better clarify the novelty aspects of their paper concerning the current state of the art?

Authors should point out their conclusion and highlight the main contribute of this study to the current knowledge. The novelty obtained and the economical impact of the results obtained in this study should be clarified and stressed.

Below are some of the ambiguities that need to be clarified, questions that should be answered and changes that should be made in the manuscript. Thus, I believe the paper is worthy of publishing after major revision.

The “Introduction” section would need some revision about grammar and style.

Several comments are listed below:

Page 2, line 38: I would suggest switching the word “majority” with “majorities”.

Page 2, lines 47-50: this sentence sounds excessively long, please edit it.

Page 2, lines 51-53: this sentence sounds redundant, I would suggest authors shorten it.

Page 2, line 53: I would suggest to modify “Authors” into “Several authors”.

Page 2, line 54: I would suggest removing “etc”.

Page 2, lines 55-58: this sentence sounds excessively long, please edit it.

Page 2, lines 64-66: this sentence appears excessively long and shows some clarity issues, please edit it.

Page 3 line 99: Please report the strain number and the reference.

Page 3 line 106: Please specify which culture medium was used in the experiments.

Page 4 line 125: Please specify which wavelength it was adopted to measure the algal growth.

Bibliography

Acien G, Hall D, Guerrero E, Rao K, Molina-Grima E (2003). Outdoor production of Phaeodactylum tricornutum biomass in a helical reactor. Journal of biotechnology. 103. 137-52. 10.1016/S0168-1656(03)00101-9.

Benavides AMS, Torzillo G, Kopecký J, Masojídek J (2013) Productivity and biochemical composition of Phaeodactylum tricornutum (Bacillariophyceae) cultures grown outdoors in tubular photobioreactors and open ponds. Biomass Bioenergy 54, 115–122.

Branco-Vieira M, San Martin S, Agurto C, Santos MA, Freitas MAV, Mata TM, Martins AA, Caetano NS (2018) Potential of Phaeodactylum tricornutum for Biodiesel Production under Natural Conditions in Chile. Energies 2018, 11, 54.

Christensen P, Peng G, Vogel F, Iversen B (2014). Hydrothermal Liquefaction of the Microalgae Phaeodactylum tricornutum: Impact of Reaction Conditions on Product and Elemental Distribution. Energy & Fuels. 28. 5792–5803. 10.1021/ef5012808.

Fernández FGA; Pérez JAS, Sevilla JMF, Camacho FG, Grima EM (2000) Modeling of eicosapentaenoic acid (EPA) production from Phaeodactylum tricornutum cultures in tubular photobioreactors. Effects of dilution rate, tube diameter, and solar irradiance. Biotechnol. Bioeng., 68, 173–183.

Gao B, Chen A, Zhang W, Li A, Zhang C (2017) Co-production of lipids, eicosapentaenoic acid, fucoxanthin, and chrysolaminarin by Phaeodactylum tricornutum cultured in a flat-plate photobioreactor under varying nitrogen conditions. J. Ocean Univ. China, 16, 916–924.

Quelhas PM, Trovão M, Silva JT et al. (2019). Industrial production of Phaeodactylum tricornutum for CO2 mitigation: biomass productivity and photosynthetic efficiency using photobioreactors of different volumes. J Appl Phycol 31, 2187–2196 https://doi.org/10.1007/s10811-019-1750-0.

Round 2

Reviewer 2 Report

The authors took the reviewer’s comments on board and improved the manuscript “Scale-up cultivation of Phaeodactylum tricornutum to produce biocrude by hydrothermal liquefaction”. However, there are additional issues that need further clarification.

For example, after suggestion, the authors specified which culture medium was used in the experiments. However, in the manuscript was not reported which kind of nitrate compound is used in this medium (i.e. sodium nitrate).

In my opinion, it would have been important to give this information because other authors as Huete‑Ortega et al., 2018 investigated the effect of a reduced source of nitrogen (i.e. ammonium) for the same strain.

Could you explain why did you choose this medium? Could authors provide the chemical composition of the medium?

Moreover, in the same manuscript mentioned above (Huete‑Ortega et al., 2018) was investigated the combined effect of a reduced source of nitrogen and of high light (1000 μmol m−2 s−1). Could you explain why you didn't take into account this information?

Taking into account the relevance of the topic focused on this manuscript, I think that it can be accepted for publication after minor revisions.

Page 1, line 19: is it correct that the only nutrient that was varied in the culture medium was the nitrate? Please, change the sentence.

Page 1, line 21: is it correct that the only nitrate that was present into the medium was the sodium nitrate? Please, change the sentence.

Page 2, line 51: I would suggest adding the article “the” before the word “manipulation”.

Page 2, line 60: I would suggest to modify “produce” into “produces”.

Page 2, line 70: I would suggest removing “a” before “lipid content”.

Page 3 line 88: I would suggest to modify “use” into “used”.

Page 3 lines 90-91: maybe the authors mean "physiological parameters" and not “different variables in the culture media”.

Page 3 line 110: this sentence shows some clarity issues, please edit it. I would suggest to modify “specie” into “species”.

Page 3 line 108: is 11.8 mM the standard concentration of the medium employed in this study?

Page 3, paragraph 2.3. “Scale-up cultivation trials”: could authors provide more information about the bubble column photobioreactor (e.g. height, diameter, type of material such as glass, PVC or other)? Could authors indicate how many replicates of the experiment were performed?

Page 3 line 108: please, clarify if the effects of different nitrogen concentration were evaluated both in absence and presence of additional CO2.

Page 3 line 129: [the supernatant from a 1 mL centrifuged culture sample] I would suggest to change this sentence into “of 1 mL of supernatant sample of a centrifuged culture”. Concerning this point, could author provide more information about centrifuge rpm and duration? Was nitrate concentration measured on the centrifuged sample as is or was some treatment applied before?

Page 5, paragraph 2.6. “Statistical analysis”: why didn’t authors consider a multivariate statistics of principal component to evaluate which variables were more discriminant for the results obtained?

Page 6, paragraph 2.3. “Results and Discussion”: Please, could authors report if the scatter bars on the figures refer to standard error or standard deviation? Please indicate the same also in the data showed in the tables. In this regard, please indicate also the n of the sample.

Page 20 line 600-604: considering that the lower biomass productivity obtained in the scale-up experiment represents a weakness on the production process, how would authors explain the economic advantages of similar plants for biofuel production? Could authors provide a cost-benefit analysis of the process to justify their conclusions?

Bibliography

Huete‑Ortega et al. Biotechnol Biofuels (2018) 11:60 https://doi.org/10.1186/s13068-018-1061-8
